# Prevalence of Poor Sleep Quality and Associated Factors in Individuals with Rheumatoid Arthritis: A Cross-Sectional Study

**DOI:** 10.3390/medicina59091633

**Published:** 2023-09-08

**Authors:** Isela Esther Juárez-Rojop, Ana Fresán, Alma Delia Genis-Mendoza, Carolina Cerino-Palomino, German Alberto Nolasco-Rosales, Thelma Beatriz González-Castro, María Lilia López-Narváez, Francisco Olan, Mario Villar-Soto, Carlos Alfonso Tovilla-Zárate, Humberto Nicolini

**Affiliations:** 1División Académica de Ciencias de la Salud, Universidad Juárez Autónoma de Tabasco, Villahermosa 86100, Mexico; iselajuarezrojop@hotmail.com (I.E.J.-R.); caro18cp@gmail.com (C.C.-P.); ganr_1277@live.com.mx (G.A.N.-R.); 2Subdirección de Investigaciones Clínicas, Instituto Nacional de Psiquiatría Ramón de la Fuente Muñiz, Ciudad de México 14370, Mexico; a_fresan@yahoo.com.mx; 3Servicio de Atención Psiquiátrica, Hospital Psiquiátrico Infantil Dr. Juan n. Navarro, Ciudad de México 14080, Mexico; adgenis@inmegen.gob.mx; 4Laboratorio de Genómica de Enfermedades Psiquiátricas y Neurodegenerativas, Instituto Nacional de Medicina Genómica, Ciudad de México 14610, Mexico; 5División Académica Multidisciplinaria de Jalpa de Méndez, Universidad Juárez Autónoma de Tabasco, Jalpa de Méndez 86205, Mexico; thelma.glez.castro@gmail.com; 6División Académica Multidisciplinaria de Comalcalco, Universidad Juárez Autónoma de Tabasco, Comalcalco 86650, Mexico; dralilialonar@yahoo.com.mx; 7Hospital de alta Especialidad “Gustavo A. Rovirosa Pérez”, Secretaría de Salud, Villahermosa 86280, Mexico; adgenis76@gmail.com (F.O.); mariovillarsoto@hotmail.com (M.V.-S.)

**Keywords:** rheumatoid arthritis, depression, anxiety, Mexican population

## Abstract

*Background and Objectives*: Poor sleep quality has been frequently observed in individuals with rheumatoid arthritis. In the present study, we analyzed the presence of poor sleep quality in a sample of Mexican individuals with rheumatoid arthritis; then, we compared sociodemographic and clinical characteristics among patients to determine risk factors for poor sleep quality. *Materials and Methods*: In this cross-sectional study, we included 102 individuals with rheumatoid arthritis from a hospital in Mexico. We evaluated disease activity (DAS28), quality of sleep using the Pittsburgh Sleep Quality Index, and the presence of depression and anxiety with the Hospital Anxiety and Depression Scale. We performed a Chi-square test and a *t*-test. Then, we performed a logistic regressions model of the associated features in a univariable analysis. *Results*: Poor sleep quality was observed in 41.75% of the individuals with rheumatoid arthritis. Being married was a proactive factor (OR 0.04, 95% CI 0.1–0.9, *p* = 0.04), whereas having one’s hips affected or presenting with anxiety and depression was associated with poor sleep quality (OR 4.6, 95% CI 1.2–17.69, *p* = 0.02). After a multivariate analysis, having anxiety (OR 5.0, 95% CI 1.4–17.7, *p* < 0.01) and depression (OR 9.2, 95% CI 1.0–8.1, *p* < 0.01) remained associated with a higher risk of having poor sleep quality. Other clinical characteristics among patients were not significantly different. *Conclusions*: Our results showed that individuals with rheumatoid arthritis who also presented with depression or anxiety had a higher risk of suffering from poor sleep quality. However, more studies with larger samples are necessary to replicate these results in the Mexican population.

## 1. Introduction

Rheumatoid arthritis (RA) is a chronic, systemic, inflammatory disorder of unknown cause; it is characterized by symmetrical, polyarticular pain and swelling, morning stiffness, and fatigue [1]. The global prevalence of rheumatoid arthritis is 0.1% to 1% and it is more common in developed countries [2,3,4]. However, in Mexico, rheumatoid arthritis prevalence reaches 2.8% [5].

Manifestations of rheumatoid arthritis are not limited to joint stiffness and pain; they could affect other organs and cause systemic effects such as pulmonary fibrosis, neuropathy, and pericarditis [6]. In addition, patients with this disease have several factors that may contribute to RA’s manifestations, including disorders such as psychological problems or sleep disturbances [6,7] such as difficulty falling asleep, poor-quality sleep, non-restorative sleep, wakefulness, awakening during the night, and excessive daytime sleepiness [3,8]. Several studies have shown that sleep disturbances appear in 54% to 70% of individuals with rheumatoid arthritis; therefore, problems related to sleep quality are an important issue among these patients [9,10]. Previous meta-analyses investigating sleeping patients with rheumatoid arthritis, including studies in Turkey, Korea, China, the USA, Malaysia, and Sweden, found an increased risk of sleep disturbances among people with rheumatoid arthritis [7,11]. 

Sleep disturbances in those with RA worsen their quality of life and impair their daily activity [12]. Sleep problems increase with disease duration, but reductions in sleep quality relate more to pain and functional impairment [13]. Possible causes of sleep disturbances in RA include inflammatory disease activity, joint pain, sleep apnea, restless leg syndrome, fatigue, a decreased quality of life, and psychological disorders such as depression [7,10,14]. The prevalence of depression in individuals with rheumatoid arthritis ranges between 14% and 48%. It is known that these patients have poorer long-term outcomes including increased pain, more comorbidities, and increased mortality levels [15,16]. In addition, pain, fatigue, and disability in those with RA cause mood changes related to anxiety and depression. Furthermore, impaired coping responses to pain, fatigue, and disability in patients with depression reduce physical exercise and social interaction, leading to poorer RA outcomes [16]. Although some studies found that features like marital status, gender, and socioeconomic level can cause mood disorders in RA, other reports failed to establish such associations [17,18,19]. On the other hand, factors for optimal sleep for those with RA are a lower body mass index, better quality of life, treatment, and disease remission [14,20,21].

However, at the present time, there are no reports of Mexican individuals with rheumatoid arthritis evaluating the prevalence of sleep deprivation or associated factors relating to quality of sleep. Therefore, the purpose for this study was (1) to determine the prevalence of poor sleep quality and (2) to determine the possible risk factors associated with poor sleep quality in individuals with rheumatoid arthritis. We explored the possibility that RA activity is involved in sleep disturbances in patients with RA and whether symptoms of depression and anxiety worsen sleep quality in these subjects.

## 2. Materials and Methods

### 2.1. Sample

We recruited subjects who attended their programmed consultation at the outpatient Rheumatology Service of the “Hospital Regional de Alta Especialidad Dr. Gustavo A. Rovirosa Pérez” in Tabasco, Mexico. Inclusion criteria were patients with an RA diagnosis, 18–85 years old, and subjects with a recent C-reactive protein (CRP) determination. We excluded patients with a previous diagnosis of depression or anxiety. The diagnosis of rheumatoid arthritis in every individual was confirmed by a rheumatologist following the 2010 Rheumatoid Arthritis Classification Criteria (from the American College of Rheumatology in collaboration with the European League Against Rheumatism) [22]. A total of 102 individuals were recruited.

### 2.2. Demographic and Clinical Features

Demographic variables such as sex, age, years of education, and current marital and employment statuses were gathered in a format previously designed for the present study. Clinical variables such as years of illness evolution; main affected joints; other medical comorbidities; use of alcohol, tobacco and cannabis; as well as current pharmacological and physical treatments were obtained from clinical records to avoid possible memory bias.

### 2.3. Disease Activity Score

To measure RA disease activity, the 28-joint Disease Activity Score (DAS28) was used. The attending rheumatologist calculated the DAS28 score of each patient using the DAWN Visual DAS28 Calculator [23]. Specific swollen joints, tender joints, and CRP levels were used for the calculations [24,25]. We used CRP levels previously measured and filed for each individual, considering the most recent evaluation. The DAS28 score was used to classify disease activity; values ≥ 2.6 were considered as active disease while lower values were defined as disease remission.

### 2.4. Hospital Anxiety and Depression Scale

Depression and anxiety symptoms were assessed using the Hospital Anxiety and Depression Scale (HADS). The HADS is validated in Spanish [26]. It is an instrument of 14 items (seven for anxiety and seven for depression) scored on a Likert scale from 0 (lowest severity) to 3 (highest severity). The results from each scale ranged from 0 to 21. Two graduated physicians applied the HADS to all included patients. A cutoff point of ≥8 defines a positive screening for significant symptoms of anxiety or depression [27,28]. 

### 2.5. Pittsburgh Sleep Quality Index

The quality of sleep was evaluated using the Pittsburgh Sleep Quality Index (PSQI) [29]. We used a validated Spanish version [30]. This questionnaire provides a global evaluation of the quality of sleep by assessing seven hypothetical components: subjective sleep quality, sleep latency, sleep duration, habitual sleep efficiency, sleep disturbances, use of sleeping medication, and daytime dysfunction. Each component is graded from 0 to 3, and a final score is obtained. A total score of ≥5 is indicative of poor sleep quality. It was used as a cutoff point to divide the sample into patients with adequate sleep (Adequate-SQ) and patients with poor sleep quality (Poor-SQ).

### 2.6. Statistical Analysis

Demographic and clinical characteristics are presented as frequencies and percentages for categorical variables while continuous variables are presented as means and standard deviations (SDs). First, all demographic and clinical variables were compared between patients with adequate sleep and those with a poor quality of sleep. Chi-square tests (Fisher exact test) with an estimated odds ratio (95% CI) were used to compare categorical variables while independent samples’ t-tests were used for continuous variables. An effect size for t-tests (Cohen *d*) was computed for significant results from the comparative analyses. Effect sizes were interpreted as small (*d* = 0.2), medium (*d* = 0.5), and large (*d* = 0.8) [31]. Variables with significant differences between groups in the comparative analyses were included in a multivariate logistic regression analysis to determine which variables conferred an important risk for poor sleep quality. The significance for all tests was established at *p* ≤ 0.05; all tests were performed with the SPSS version 22 for Windows, PC.

### 2.7. Ethical Considerations

This study was approved by the Ethical Committee of the “*Universidad Juárez Autónoma de Tabasco*” (UJAT-IB-2018-04) in compliance with ethical principles and guidelines for the protection of human individuals under research. All participants signed a written, informed consent form after they were given verbal and written explanations of the research objectives and procedures. They did not receive any economical remuneration for their participation.

## 3. Results

Of the 102 individuals with rheumatoid arthritis included, 93.1% (n = 95) were women with a mean age of 52.0 years (SD = 12.6, range 21–78 years) and an average length of education of 6.4 years (SD = 4.2, range 0–19 years). More than half were married (67.6%, n = 69) and were engaged in non-remunerated activities (housewife 83.3%, n = 85; student 2.9%, n = 3; unemployed 2.0%, n = 2).

The mean age of illness onset was 44.1 years (SD = 11.9, range 16–69), with an illness evolution of 7.9 years (SD = 5.4, range 1–42 years). Hands (73.5%, n = 75) were the most frequently reported area affected by RA, followed by knees (45.1%, n = 46), ankles and feet (26.5%, n = 27), hips (13.7%, n = 14), and the neck (4.9%, n = 5). According to the DAS28 total scores (mean score 2.7, SD = 2.7, range 0.1–7.3), 53.9% (n = 55) of the individuals were in disease remission (DAS28 < 2.6). All the patients were undergoing pharmacological treatment and only 11.8% (n = 12) reported receiving physiotherapy. Less than 10% of the sample reported substance use and the main medical comorbidity reported was systemic arterial hypertension (25.5%, n = 26). Significant depressive symptoms were present in 15.7% (n = 16) of the individuals while anxiety symptoms were reported in 25.5% (n = 26) of the patients.

According to the proposed cutoff point of the PSQI, almost half of the sample (49%, n = 50) had poor sleep quality. The demographic and clinical characteristics by current sleep quality are shown in Table 1. As can be seen, a higher proportion of patients with poor sleep quality reported significantly affected hips and symptoms of anxiety and depression, while those patients who were married reported adequate sleep quality more frequently than those who were not married.

The PSQI components were compared between individuals with and without affected hips and symptoms of anxiety and depression, as well as those who were single or married. The subjective sleep quality, sleep latency, sleep disturbances, and daytime dysfunction areas were more affected in those with depressive and anxious symptoms. Only sleep latency differed between married and single participants. Sleep disturbances were more prevalent in those whose hips were affected (Table 2). All differences showed a moderate to high size effect.

The multivariate logistic regression analysis evaluated the involvement of hips, symptoms of anxiety, depressive symptoms, and marital status. This evaluation showed that both depression and anxiety were important predictors for poor sleep quality in patients with rheumatoid arthritis (Table 3). After adjustments, this final logistic regression equation correctly classified 71.6% of the cases and was significant for our sample according to the Hosmer and Lemeshow statistical value (*p* = 0.73).

## 4. Discussion

In the present study, we observed an association between depressive and anxiety symptoms with subjective sleep quality, sleep latency, sleep disturbances, and daytime disfunction (in the PSQI). Likewise, hip involvement increased the sleep disturbances’ component and being married reduced the sleep latency component; nonetheless, these associations were not significant in our logistic regression. To our knowledge, there are no previous studies that analyzed these associations in a Mexican population.

We found that 49% of our sample had poor sleep quality. This result is similar to poor sleep frequencies reported in previous studies that evaluated rheumatoid arthritis [3,9,10]. Additionally, it was observed that RA patients have a high prevalence of sleep disorders when compared with controls [3,32].

In our study, individuals who were married showed a better sleep quality than those who were single, as they had shorter sleep latencies. It was reported that, in older adults, sleep quality measured by actigraphy was enhanced when they slept with their partners; nevertheless, subjective sleep quality was not affected by sleeping alone [33]. Another report showed that elder RA patients had lower PSQI scores when they were married, but younger patients did not show a similar association [10]. Nonetheless, being married was observed to be associated with a better sleep quality in younger couples [34]. Relationship quality could improve or impair sleep quality, as couples with fewer partner problems had less sleeping trouble [35]. Sleeping with a partner has many individual variables involved affecting both individuals; mood disorders are among them [36]. As our analysis only took data from one individual (the RA patient) in the marriage, these factors were not reflected in our results; in fact, the association between sleep quality and being married was not present in our logistic regression model.

In our sample, patients whose hips were affected showed an impaired sleep quality. This impairment was associated with increased sleep disturbances; this could be interpreted as a manifestation of enhanced pain perception that impaired their sleep quality. The PSQI evaluates sleep disturbances by asking how frequently sleep is disturbed by specific nuisances; pain is among these disturbances. Hip involvement increases its prevalence over time [37]. This suggests that some patients develop additional painful articulations as the disease progresses. Pain has a complex relationship with sleep quality. It was reported that pain in RA patients is associated with a worse sleep quality [3,7,10]. Pain also has an indirect association with sleep impairment by inducing a depressed mood [21]. Poorer sleep quality and higher pain intensity also were reported in chronic pain unrelated to cancer in Mexican patients [38,39]. Pain produces nightly awakenings, sleep fragmentation, and reduced sleep efficiency [21,32], which can be detected as sleep disruption in the PSQI. On the other hand, impaired sleep can enhance pain perception [21]. Unfortunately, we did not use pain scales with our participants; therefore, there was not a reliable way to measure pain. Since a hip involvement–impaired sleep association was not present in the logistic regression model, we can suppose that this association is indirectly related to pain.

The strongest associations were found with depression and anxiety. These associations remained through all our analyses. We found high frequencies of depression and anxiety among our patients compared with the mean prevalence in Mexico [40]. The frequency of mood disorders in our sample was similar to those reported in RA patients in other countries [16,41]. Comparing patients with poor and adequate sleep quality, we found even higher frequencies of depression and anxiety in those with impaired sleep. Mood disorders and sleep quality were already observed in RA patients [3,9,42]; in our sample, for instance, items in the PSQI affected by depression and anxiety were subjective to sleep quality, sleep latency, sleep disturbances, and daytime dysfunction. Additionally, when RA patients were compared with healthy individuals, similar associations were observed: worse subjective sleep quality, sleep latency, habitual sleep efficiency, and sleep disturbances in individuals with RA [8]. In the logistic regression model, poor sleepers showed a 9.2-fold probability of being depressed and 5.0 times probability of being anxious. It is known that sleep disorders increase the risk for depression [43,44]; likewise, depression risk is increased with impaired sleep quality [45,46]. The majority of anxiety disorders are related to sleep disturbances; however, the causal relationship is not clear [47]. RA is also associated with depression and anxiety [41,48]. Sleep disturbances have been suggested as links between RA and mood disorders, each one worsening joint symptoms [21,49]. It was suggested that RA could enhance these associations through increased inflammatory cytokines’ levels (IL-1, IL-6, and TNF-α) as a consequence of a dysregulated immune system [16,49]. Finally, it was also suggested that chronic inflammation, chronic pain, disordered sleep, and psychological stress all together make individuals with RA highly vulnerable to sleep disorders, mood disorders, and disease flares [50].

Sleep is a complex and multifactorial process, and RA patients have many vulnerabilities. It is important that general practitioners and rheumatologists are aware that symptoms and signs like poor sleep quality, pain, fatigue, and uncontrolled disease activity increase the risk of mood disorders in RA patients [16,20]. In the same way, mood disorders can worsen RA disease activity. Early detection of mood disorders could ease RA disease activity control [51].

The present study has some limitations consider. The sample size was small, and we only evaluated subjects from a single referral center. Also, there was a high female/male ratio (14:1). Although RA is more frequently observed in women than in men in the Mexican population (6:1) [52,53], our sample could not be representative for male RA patients. The study design was cross-sectional, so it was not possible to establish causal relationships or see changes in related factors over time. Sleep quality was evaluated with the PSQI; therefore, these results just evaluated subjective sleep quality. Finally, we did not assess whether patients were receiving psychotherapy or counseling outside the hospital.

## 5. Conclusions

In conclusion, this study shows that depression and anxiety were associated with impaired sleep quality in 49% of our sample of individuals with RA from a Mexican population. We associated depressive and anxiety symptoms with subjective sleep quality, sleep latency, sleep disturbances, and daytime dysfunction. Although hip involvement increased the sleep disturbances’ component and being married reduced the sleep latency component in our univariable analysis, they were not significant in our logistic regression. It is necessary to perform other studies with larger samples as well longitudinal studies in order to determine possible causal relationships.

## Figures and Tables

**Table 1 medicina-59-01633-t001:** Demographic and clinical characteristics according to sleep quality.

	Total Sample n = 102	Adequate SQ n = 52	Poor SQ n = 50	Statistic
Demographic Features n %				
Gender—Women	95 93.1	49 94.2	46 92.0	Fisher = 0.7, OR = 0.7 95% CI = 0.1–3.3
Age *	52.0 12.6	52.3 12.5	51.7 12.8	t = 0.24, *p* = 0.81
Education (years) *	6.4 4.2	6.8 3.5	6.0 4.9	t = 0.97, *p* = 0.33
Marital status—Married	69 67.6	40 76.9	29 58.0	**Fisher = 0.04, OR = 0.4** **95% CI = 0.1–0.9**
Occupation—Non-remunerated	90 88.2	46 88.5	44 88.0	Fisher = 1.0, OR = 1.0 95% CI = 0.3–3.4
Medical Comorbidities n %				
Diabetes—Yes	17 16.7	5 9.6	12 24.0	Fisher = 0.06, OR = 2.9 95% CI = 0.9–9.1
Arterial hypertension—Yes	26 25.5	9 17.3	17 34.0	Fisher = 0.07, OR = 2.4 95% CI = 0.9–6.2
Liver disease—Yes	2 2.0	-	2 4.0	Fisher = 0.2
Current Substance Use n %				
Alcohol–Yes	9 8.8	6 11.5	3 6.0	Fisher = 0.4, OR = 0.4 95% CI = 0.1–2.0
Tobacco–Yes	5 4.9	2 3.8	3 6.0	Fisher = 0.6, OR = 1.5 95% CI = 0.2–9.9
Marihuana—Yes	1 1.0	1 1.9	-	Fisher = 1.0
Rheumatoid Arthritis Features n %				
Age of onset *	44.1 11.9	44.5 12.2	43.7 11.7	t = 0.31, *p* = 0.75
Years of evolution *	7.9 5.4	7.8 6.2	7.9 4.4	t = −0.14, *p* = 0.88
Disease activity (DAS28)—Yes	47 46.1	20 38.5	27 54.0	Fisher = 0.1, OR = 1.8 95% CI = 0.8–4.1
Affected joint				
Hands	75 73.5	38 73.1	37 74.0	Fisher = 1.0, OR = 1.0 95% CI = 0.4–2.5
Knees	46 45.1	20 38.5	26 52.0	Fisher = 0.2, OR = 1.7 95% CI = 0.7–3.8
Hips	14 13.7	3 5.8	11 22.0	**Fisher = 0.02, OR = 4.6** **95% CI = 1.2–17.6**
Neck	5 4.9	-	5 10.0	Fisher = 0.05
Ankles and Feet	27 26.5	14 26.9	13 26.0	Fisher = 1.0, OR = 0.9 95% CI = 0.3–2.3
Physiotherapy—Yes	12 11.8	5 9.6	7 14.0	Fisher = 0.5, OR = 1.5 95% CI = 0.4–5.1
Mental Health n %				
Anxiety—Yes	26 25.5	4 7.7	22 44.0	**Fisher < 0.001, OR = 9.4** **95% CI = 2.9–30.1**
Depression—Yes	16 15.7	1 1.9	15 30.0	**Fisher < 0.001, OR = 21.8** **95% CI = 2.7–173.1**

* Data presented in means and SDs. Significance is presented in bold.

**Table 2 medicina-59-01633-t002:** PSQI components according to demographic and clinical features that differed between individuals with adequate and poor sleep quality.

	Marital Status		Hip Involvement		Depression		Anxiety	
	Single n = 33	Married n = 69	No n = 88	Yes n = 14	No n = 86	Yes n = 16	No n = 76	Yes n = 26
Subjective sleep quality	0.9 (1.0)	0.8 (0.8)	0.7 (0.8)	1.2 (0.9)	0.7 (0.8)	1.5 (1.0)	0.6 (0.8)	1.3 (0.9)
	*p* = 0.47		*p* = 0.09		*p* = 0.001; *d* = 0.8		*p* = 0.001; *d* = 0.8	
Sleep latency	1.5 (1.0)	1.0 (1.0)	1.1 (1.0)	1.5 (1.2)	1.0 (1.0)	2.1 (0.7)	0.9 (1.0)	1.8 (1.0)
	*p* = 0.01; *d* = 0.5		*p* = 0.23		*p* < 0.001; *d* = 1.2		*p* < 0.001; *d* = 0.9	
Sleep duration	0.5 (0.7)	0.6 (0.8)	0.6 (0.8)	0.7 (0.8)	0.5 (0.8)	0.8 (0.9)	0.5 (0.7)	0.8 (1.0)
	*p* = 0.66		*p* = 0.43		*p* = 0.32		*p* = 0.12	
Habitual sleep efficiency	0.6 (1.0)	0.5 (0.9)	1.0 (0.1)	0.9 (0.2)	0.5 (0.9)	0.8 (1.1)	0.5 (0.9)	0.8 (1.0)
	*p* = 0.63		*p* = 0.64		*p* = 0.35		*p* = 0.09	
Sleep disturbances	1.3 (0.6)	1.0 (0.6)	1.0 (0.6)	1.5 (0.6)	1.0 (0.6)	1.7 (0.5)	1.0 (0.6)	1.5 (0.6)
	*p* = 0.11		*p* = 0.04; *d* = 0.8		*p* < 0.001; *d* = 1.2		*p* < 0.001; *d* = 0.8	
Sleeping medication	0.3 (0.8)	0.06 (0.3)	0.1 (0.5)	0.2 (0.8)	0.1 (0.5)	0.2 (0.7)	0.08 (0.3)	0.3 (0.9)
	*p* = 0.09		*p* = 0.35		*p* = 0.46		*p* = 0.18	
Daytime dysfunction	0.7 (0.9)	0.6 (0.8)	0.6 (0.8)	0.9 (1.1)	0.5 (0.6)	1.5 (1.1)	0.5 (0.7)	1.1 (0.9)
	*p* = 0.46		*p* = 0.22		*p* = 0.001; *d* = 1.1		*p* = 0.001; *d* = 0.7	

Data presented in means and SDs.

**Table 3 medicina-59-01633-t003:** Logistic regression model for poor sleep quality in individuals with RA.

	β	Odds Ratio	95% CI	*p*
Initial model				
Marital status—Married	−0.6	0.5	0.1–1.3	0.17
Anxiety—Yes	1.7	5.5	1.5–19.7	0.008
Depression—Yes	2.2	9.1	0.9–83.7	0.051
Hip involvement—Yes	1.2	3.5	0.7–16.2	0.10
Final model				
Anxiety—Yes	1.6	5.0	1.4–17.7	0.01
Depression—Yes	2.2	9.2	1.0–81.6	0.04

## Data Availability

Data available upon request.

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
