# Peer review of "Prevalence of Poor Sleep Quality and Associated Factors in Individuals with Rheumatoid Arthritis: A Cross-Sectional Study"

_medicina, 2023, doi:10.3390/medicina59091633_

Round 1

Reviewer 1 Report

article on Prevalence of poor sleep quality and associated factors in individuals with rheumatoid arthritis: A cross-sectional study

The topics chosen are quite interesting and up to date.

the abstract is prepared according to scientific principles, which consists of background, method, results, conclusions and keywords, suggestions: for an abstract, please explain briefly about the research design, samples, variables, instruments and data analysis at the point of the method.

an introductory chapter that contains background, please emphasize what phenomena you encountered so that it becomes your research topic, of course it is supported by relevant data

the research method is good, what about the inclusion and exclusion criteria along with how you choose the sample and how about the ethical test that you carried out? Please explain

the results and discussion are quite good, add your opinion to the discussion which is supported by theory

conclusions must answer the research objectives

references are used accordingly

Author Response

Reviewer #1

The topics chosen are quite interesting and up to date.

The abstract is prepared according to scientific principles, which consists of background, method, results, conclusions and keywords, suggestions: for an abstract, please explain briefly about the research design, samples, variables, instruments and data analysis at the point of the method.

Thank you for your observation. We modified the methods in the abstract.

Changes in manuscript:

Page 1. Line 28 – 33. Materials and Methods. In this cross-sectional study, we included 102 individuals with rheumatoid arthritis from a hospital in Mexico. We evaluated disease activity (DAS28), quality of sleep using the Pittsburgh Sleep Quality Index, and the presence of depression and anxiety with the Hospital Anxiety and Depression Scale. We performed Chi-square test and t-test. Then, we performed a logistic regressions model of the associated features in the univariable analysis.

An introductory chapter that contains background, please emphasize what phenomena you encountered so that it becomes your research topic, of course it is supported by relevant data.

We appreciate your comments. We added information in Introduction section about the relationship between rheumatoid arthritis, depression, anxiety, and sleep quality.

Changes in manuscript:

Page 2, Lines59 to 61. Previous meta-analysis investigating sleep patients with rheumatoid arthritis, including studies of Turkey, Korean, China, USA, Malaysia, and Swedish, encountered an increased risk of sleep disturbances amount people with rheumatoid arthritis [7,11].

Page 2, Line 63 to 67 Sleep disturbances in RA worse the quality of life and impairs the daily activity [12]. Sleep problems increase with disease duration, but reductions in sleep quality relates more with pain and functional impairment [13]. Possible causes of sleep disturbances in RA include inflammatory disease activity, joint pain, sleep apnea, restless leg syndrome, fatigue, decreased quality of life and psychological disorders such as depression [7,10,14].

Page 2, Lines 70- 77. In addition, pain, fatigue and disability in RA causes mood changes as anxiety and depression Furthermore, impaired coping responses to pain, fatigue and disability in patients with depression reduces physical exercise and social interaction, leading to poorer RA outcomes [16]. Although some studies found that features like marital status, gender and socioeconomic level can impair mood disorders in RA, other reports fail to establish such associations [17-19]. On the other hand, factors for optimal sleep in RA are lower body mass index, better quality of life, treatment, and disease remission [14,20,21].

The research method is good, what about the inclusion and exclusion criteria along with how you choose the sample and how about the ethical test that you carried out? Please explain.

Thank you for your observation. We explained the inclusion and exclusion criteria, and how we recruited the patients from the hospital. We also moved the Ethical considerations to the end of the Methods section and included number of approval of the study.

Changes in manuscript:

Page 2, Lines 87-91. Inclusion criteria were patients with RA diagnosis, 18–85 years old, and subjects with a recent C-reactive protein (CRP) determination. We exclude patients with previous diagnosis of depression or anxiety.

Page 3, Line 148 -154. Ethical considerations.

This study was approved by the Ethical Committee of the “Universidad Juárez Autónoma de Tabasco” (UJAT-IB-2018-04) in compliance with ethical principles and guidelines for the protection of human individuals under research. All participants signed a written informed consent after they were given verbal and written explanations of the re-search objectives and procedures. They did not receive any economical remuneration for their participation.

The results and discussion are quite good, add your opinion to the discussion which is supported by theory.

We agree with this observation. We moved our opinion from the conclusion to discussion and added citations to support it.

Changes in manuscript:

Page 8, Lines 268-273. Sleep is a complex and multifactorial process, and RA patients have many vulnerabilities. It is important that general practitioners and rheumatologist are aware that symptoms and signs like poor sleep quality, pain, fatigue, and uncontrolled disease activity increase the risk of mood disorders in RA patients [16,20]. In the same way, mood disorders can worsen RA disease activity. An early detection of mood disorders could ease RA disease activity control [48].

Conclusions must answer the research objectives.

We appreciate your comment. We added the main findings of our research in the Conclusions section.

Changes in manuscript:

Page 8, Lines 284-291. We associated depressive and anxiety symptoms with subjective sleep quality, sleep latency, sleep disturbances and daytime disfunction. Although hip affectation increased sleep disturbances component and being married reduced the sleep latency in univariable analysis, they were not significant in logistic regression.

References are used accordingly.

Thank you.

Reviewer 2 Report

good paper, well writen,

some old rererences m7s5 be updated

Specific comments:

The main question addressed by the research  is the poor sleep quality in a sample of Mexican individuals with rheumatoid arthritis asdociated to sociodemographic and clinical characteristic which was relevant and interesting.

The originality is fair, the paper is well written, the text clear and easy to read, the conclusions are consistent with the evidence and arguments presented they address the main question posed. The limitation of the study is the small sample.

Author Response

Reviewer #2.

Good paper, well written

Some old references must be updated.

We updated the outdated references.

Specific comments:

The main question addressed by the research is the poor sleep quality in a sample of Mexican individuals with rheumatoid arthritis associated to sociodemographic and clinical characteristic which was relevant and interesting. The originality is fair, the paper is well written, the text clear and easy to read, the conclusions are consistent with the evidence and arguments presented they address the main question posed. The limitation of the study is the small sample.

Thank you for your kind observations. We elaborated further about the limitations of our study and emphasized about the sample size.

Changes in manuscript:

Page 8, Line 274-278. The sample size was small, and we only evaluated subjects from a single referral center. Also, there was a high female/male ratio (14:1). Although RA is more frequently observed in women than in men in the Mexican population (6:1) [49,50], our sample could not be representative for male RA patients.

Reviewer 3 Report

Dear authors,

Thank you for the effort you put into your research. Your research is valuable in its subject and scope and generally well written. However, I will ask you to make some adjustments. After the edits I will give, I think your research is suitable for publication in the journal Medicina.

Specific Comments:

Introduction

This chapter is generally short, concise and clear. However, in your research, you examined the relationship of rheumatoid arthritis not only to depression and anxiety but also to many additional factors. Therefore, I suggest that you add information about how other factors (for example, marital status, socio-demographic characteristics) may have an effect with rheumatoid arthritis in a separate paragraph in this section. In particular, this information, which you will provide with the support of literature, will attract more attention of the readers.

In the last paragraph, after your purpose sentence, please give your hypothesis or hypotheses.

Methods

I recommend that you talk about the design of your research in general at the beginning of this section, and then provide ethical approval and other information.

It is useful to detail the tests as much as possible by creating separate headings for all the data you collect in the Assesment Procedure section.

Results

No revision is required in this section.

Discussion

Please begin the first part of this chapter with the major findings of your research and then continue to discuss the literature.

Best Regards.

Author Response

Reviewer #3

Dear authors,

Thank you for the effort you put into your research. Your research is valuable in its subject and scope and generally well written. However, I will ask you to make some adjustments. After the edits I will give, I think your research is suitable for publication in the journal Medicina.

Specific Comments:

Introduction

This chapter is generally short, concise and clear. However, in your research, you examined the relationship of rheumatoid arthritis not only to depression and anxiety but also to many additional factors. Therefore, I suggest that you add information about how other factors (for example, marital status, socio-demographic characteristics) may have an effect with rheumatoid arthritis in a separate paragraph in this section. In particular, this information, which you will provide with the support of literature, will attract more attention of the readers. In the last paragraph, after your purpose sentence, please give your hypothesis or hypotheses.

Thanks for your observation. We added new information regarding factors with risk and protective effects in mood and sleep in rheumatoid arthritis. We also include our hypotheses in the manuscript.

Changes in manuscript:

Page 2, Line 63 – 67. Sleep disturbances in RA worse the quality of life and impairs the daily activity [12]. Sleep problems increase with disease duration, but reductions in sleep quality relates more with pain and functional impairment [13]. Possible causes of sleep disturbances in RA include inflammatory disease activity, joint pain, sleep apnea, restless leg syndrome, fatigue, decreased quality of life and psychological disorders such as depression [7,10,14].

Page 2, Lines 70- 77. In addition, pain, fatigue and disability in RA causes mood changes as anxiety and depression. Furthermore, impaired coping responses to pain, fatigue and disability in patients with de-pression reduces physical exercise and social interaction, leading to poorer RA outcomes [16]. Although some studies found that features like marital status, gender and socioeconomic level can impair mood disorders in RA, other reports fail to establish such associations [17-19]. On the other hand, factors for optimal sleep in RA are lower body mass index, better quality of life, treatment, and disease remission [14,20,21].

Page 2, Lines 82-84. We explored the possibility that RA activity is involved in sleep disturbances in patients with AR, and whether symptoms of depression and anxiety worsen sleep quality in these subjects.

Methods

I recommend that you talk about the design of your research in general at the beginning of this section, and then provide ethical approval and other information. It is useful to detail the tests as much as possible by creating separate headings for all the data you collect in the Assessment Procedure section.

We appreciate your comments. We moved the ethical approval information to the end of the Methods sections. Likewise, we added information about how we select our subjects. We separated the Assessment section into “Demographic and clinical features”, “Disease Activity Score”, “Hospital Anxiety and Depression Scale” and “Pittsburgh Sleep Quality Index”; also, we detailed these sections.

Changes in manuscript:

Page 2, Line 87 – 91. We recruited subjects who attended their programmed consultation at the outpatient Rheumatology Service of the “Hospital Regional de Alta Especialidad Dr. Gustavo A. Rovirosa Pérez” in Tabasco, Mexico. Inclusion criteria were patients with RA diagnosis, 18–85 years old, and subjects with a recent C-reactive protein (CRP) determination. We exclude patients with previous diagnosis of depression or anxiety.

Page 2, Line 107-111. 3. Disease Activity Score. To measure RA disease activity, the 28-joint Disease Activity Score (DAS28) was used. The attending rheumatologist calculated the DAS28 score of each patient using the DAWN Visual DAS28 Calculator [23]. Specific swollen joints, tender joints, and CRP lev-els were used for the calculations [24,25].

Page 3, Lines 116 – 122. 2.4. Hospital Anxiety and Depression Scale. Depression and anxiety symptoms were assessed using the Hospital Anxiety and Depression Scale (HADS). The HADS is validated in Spanish [26], it is an instrument of 14 items (seven for anxiety and seven for depression) scored on a Likert scale from 0 (lowest severity) to 3 (highest severity).

Page 3, Lines 124-126. 2.5. Pittsburgh Sleep Quality Index. The quality of sleep was evaluated with the Pittsburgh Sleep Quality Index (PSQI) [29], we used a validated Spanish version [30].

Results

No revision is required in this section.

Thank you.

Discussion

Please begin the first part of this chapter with the major findings of your research and then continue to discuss the literature.

Thank you for your comment. We added the main findings of our study in the first paragraph of the discussion section.

Changes in manuscript:

Page7, Line 206-211.  In the present study, we observed the association between depressive and anxiety symptoms with subjective sleep quality, sleep latency, sleep disturbances and daytime disfunction (in PSQI). Likewise, hip affectation increased sleep disturbances component and being married reduced the sleep latency; nonetheless, these associations were not significant in our logistic regression. To our knowledge, there are no previous studies that have analyzed these associations in Mexican population.

Best Regards.

Reviewer 4 Report

Dear Editor,
I appreciate the opportunity to review manuscript medicina-2557099 entitled:
"Prevalence of poor sleep quality and associated factors in individuals with rheumatoid arthritis: A cross-sectional study"

I commend the authors for describing this critical and timely issue. The paper is interesting and well-written; however, I would like to highlight some issues that merit revision:

It is not detectable in the manuscript that an assessment of what might be a protective factor provided not only psychotherapy but counseling intervention, which people often use without specialist prescription to prevent or mitigate symptoms at onset. I would ask the authors whether this was evaluated during the survey and add a short paragraph on this issue; if data are unavailable, add them to the limitations.

Author Response

Reviewer #4

Dear Editor,

I appreciate the opportunity to review manuscript medicina-2557099 entitled:

"Prevalence of poor sleep quality and associated factors in individuals with rheumatoid arthritis: A cross-sectional study". I commend the authors for describing this critical and timely issue. The paper is interesting and well-written; however, I would like to highlight some issues that merit revision:

It is not detectable in the manuscript that an assessment of what might be a protective factor provided not only psychotherapy but counseling intervention, which people often use without specialist prescription to prevent or mitigate symptoms at onset. I would ask the authors whether this was evaluated during the survey and add a short paragraph on this issue; if data are unavailable, add them to the limitations.

We appreciate your observation. We only evaluated if the patients used the psychotherapy service of the hospital. As we did not ask if they used a counseling therapy out of the hospital, we included this limitation in our manuscript.

Changes in manuscript:

Page. 8, Line 280 to 282. Finally, we did not assess whether patients were receiving psychotherapy or counselling outside the hospital.

Round 2

Reviewer 3 Report

Dear authors,

manuscript is ready to publish